# COVID-19 Associated Vasculitis Confirmed by the Tissues RT-PCR: A Case Series Report

**DOI:** 10.3390/biomedicines11030870

**Published:** 2023-03-13

**Authors:** Konstantin E. Belozerov, Ilia S. Avrusin, Lyubov I. Andaryanova, Anna M. Guseva, Zaira S. Shogenova, Irina N. Belanovich, Anna V. Lobacheva, Tatiana L. Kornishina, Eugenia A. Isupova, Vera V. Masalova, Olga V. Kalashnikova, Andrey V. Nokhrin, Tatyana F. Panova, Yulia P. Dutova, Svetlana L. Myshkovskaya, Kirill Y. Kostyunin, Andrey B. Komissarov, Vyacheslav G. Chasnyk, Liudmila V. Bregel, Mikhail M. Kostik

**Affiliations:** 1Hospital Pediatry, Saint Petersburg State Pediatric Medical University, 194100 Saint Petersburg, Russia; 2Pediatric Department, Leningrad Regional Children’s Clinical Hospital, 195009 Saint Petersburg, Russia; 3Pathology Department, Irkutsk State Medical University, 664003 Irkutsk, Russia; 4Irkutsk Regional Diagnostic Centre, Department of Clinical Pathomorpholigy, 664047 Irkutsk, Russia; 5Laboratory of Molecular Virology, Smorodintsev Research Institute of Influenza, 197376 Saint Petersburg, Russia; 6Department of Pediatrics, Irkutsk State Medical Academy of Postgraduate Education, Branch of Russian Medical Academy of Continuous Professional Education, 664049 Irkutsk, Russia; 7Department of Pediatric Cardiology, Irkutsk Regional Children’s Hospital, 664022 Irkutsk, Russia

**Keywords:** COVID-19, SARS-CoV-2, real-time PCR, vasculopathy, childhood vasculitis, mesenteric thrombosis, polyarteritis nodosa, Ig A vasculopathy, Henoch–Schonlein purpura

## Abstract

Background: Several cases of skin and central nervous system vasculopathy associated with COVID-19 in children have been published, but the information is rather limited. Our study aimed to describe these cases of vasculitis associated with COVID-19 in children. Methods: In the retrospective-prospective case series study we included information regarding four children with COVID-19-associated vasculitis. In every case, we had a morphological description and the etiology was confirmed via real-time polymerase chain reaction during a tissue biopsy. Results: The most involved systems were skin (4/4), respiratory (3/4), cardiovascular (2/4), nervous (1/4), eye (1/4), kidney (1/4), and inner year (1/4). All patients had increased inflammatory markers and thrombotic parameters (D-dimer). No patient met the criteria for multisystem inflammatory syndrome in children. Two patients met polyarteritis nodosa criteria, one met Henoch–Schonlein purpura criteria, and one met unclassified vasculitis criteria. All patients were treated with systemic glucocorticosteroids (two-pulse therapy). Non-biologic DMARDs were prescribed in all cases; 1/4 patients (25%) was treated with intravenous immunoglobuline, and 3/4 (75%) were treated with biologics (etanercept, tocilizumab, and adalimumab). Conclusions: Vasculitis associated with COVID-19 could be a life-threatening condition; SARS-CoV-2 might be a new trigger or etiological agent for vasculitis and other immune-mediated diseases. Further research and collection of similar cases are required.

## 1. Introduction

COVID-19 (coronavirus disease, 2019) is a new infectious disease associated with severe acute respiratory syndrome caused by coronavirus-2 (SARS-CoV-2), which can have a fatal outcome [1]. The SARS-CoV-2 virus affects cells through angiotensin-converting enzyme-2 (ACE-2) receptors, expressed in endotheliocytes and pericytes [2,3].

After penetration, a cascade of pathophysiological reactions is launched with pro-inflammatory cytokines hyperproduction, complement activation, and enhanced interferon response, followed by microangiopathic changes similar to those in systemic lupus erythematosus [4]. Damaged endothelium and increased production of von Willebrand factor provide blood clot formation [2,3,5]. The blockade of ACE-2 receptors causes an increased level of angiotensin II, which entails a pro-inflammatory, pro-thrombotic effect and oxidative stress. Endothelial cell dysfunction and blood vessel inflammation lead to subsequent thrombosis [2,3,5].

The most common previously published clinical signs of vascular involvement are chilblains, maculopapular eruptions, erythema multiforme, purpuric and livedoid patterns, urticarial rash, and vesicular exanthema. Morphology and immunohistochemistry showed T-cell infiltrate (CD4 more than CD8), minimal B-cells, IgM, C3, C4d, and C5b-9 depositions, and fibrinogen within vessels in the derma [6]. Data regarding vasculitis in children, its clinical course, and outcomes are scarce.

Our study aimed to describe clinical patterns, treatment, and outcomes in patients with vasculitis associated with COVID-19 infection and provide an overview of previously published cases.

## 2. Materials and Methods

In the retrospective-prospective case series study, data regarding four patients (three boys and one girl) with vasculopathy associated with COVID-19 infection were included.

Inclusion criteria: (1) the presence of vasculitis, confirmed by clinical and morphological assessment; and (2) a positive SARS-CoV-2-tissue real-time polymerase chain reaction (RT-PCR) in the same biopsies.

We assessed the following for each patient to diagnose systemic vasculitis:-Epidemiology, including signs of previous COVID-19 infection (fever, cough, rhinitis, anosmia, headache, pneumonia, etc.), type of COVID-19 contact (family, class, kindergarten, or absence of evident contact), and type of COVID-19, if the corresponding assessment was performed (nasal or throat swab, PCR, IgM, or IgG against SARS-CoV-2, carried out using the enzyme-linked immunosorbent assay), in addition to the above-mentioned tissue RT-PCR;-The criteria of the World Health Organization, Centers for Disease Control and Prevention, or Royal College of Paediatrics and Child Health [7,8,9], to diagnose multisystem inflammatory syndrome in children and adolescents associated with COVID-19;-Clinical manifestations, including features of vasculitis and the following symptoms: fever, weight loss, muscle and joint pain, arthritis, rash, presence of skin or mucosal involvement, ischemia, lymphadenopathy, hepatosplenomegaly, neuropathy, and internal organ involvement (heart, lungs, brain, GI, and kidney);-Vasculitis imaging, including Doppler ultrasound, computed tomography (CT)-angiography, magnetic resonance imaging (MRI), evaluation of the caliber of involved vessels, arterial or venous thrombotic evidence, and their consequences and level of damage;-Laboratory tests, including hemoglobin, white blood cells (WBC), platelets, erythrocyte sedimentation rate (ESR), C-reactive protein (CRP), total protein, albumin, urea, creatinine, liver functional tests, ferritin, lactate dehydrogenase, coagulation status (prothrombin, fibrinogen, D-dimer), and urinalysis;-Other instrumental test results, including electrocardiogram (ECG), echocardiogram (EChoCG), abdominal ultrasound, chest CT, brain MRI, and pulmonary functional tests;-Thrombophilic genes polymorphism, which was evaluated using PCR with restriction fragment length polymorphisms according to the routine method;-Whole exome sequence to identify immunodeficiency syndromes and monogenic vasculopathies.

SARS-CoV-2 RT-PCR testing of formalin-fixed paraffin-embedded (FFPE) specimens

Nucleic acid extraction from paraffin blocks was performed using the RNeasy FFPE Kit (Qiagen, Hilden, Germany) according to the manufacturer’s instructions. Real-time RT-PCR “SARS-CoV-2 Intifica” Kits (Alkor Bio Company, Saint-Petersburg, Russia) targeting three virus genes (ORF1, ORF8, and N) were applied for SARS-CoV-2 detection.

## 3. Results

### 3.1. COVID-19 identification

Initially, COVID-19 was diagnosed in 2/4 (50%) cases: two children had signs of respiratory infection (rhinitis and cough); one patient was SARS-CoV-2 positive based on an RT-PCR test from a nasopharyngeal swab, accompanied by a positive IgG antibody, and one patient had both IgM and IgG antibodies against SARS-CoV-2. One patient had only a positive IgG against SARS-CoV-2 without signs of any other infection. Demographic, clinical, and laboratory characteristics of the included patients are shown in Table 1. All patients had positive real-time PCR in biopsy tissues on SARS-CoV-2 RNA.

### 3.2. Clinical Signs of Vasculitis

In the studied patients, the following systems were involved: skin (4/4), respiratory (3/4), cardiovascular (2/4), GI (2/4), nervous (1/4), eye (1/4), kidney (1/4), and inner ear (1/4). In all previously healthy patients, known rheumatic diseases (e.g., systemic lupus erythematosus, systemic juvenile arthritis, autoinflammatory diseases, and antiphospholipid syndrome) were ruled out, as well as known monogenic autoinflammatory disease. All patients had increased inflammatory markers (ESR and CRP) and thrombotic parameters (D-dimer). No patient met the MIS-C criteria. Two patients met the criteria of polyarteritis nodosa, one met the IgA vasculitis/Henoch–Schonlein purpura criteria, and one met the unclassified vasculitis criteria. Clinical pictures are presented in the Figure 1.

### 3.3. Treatment

Systemic glucocorticosteroids (prednisolone 1 mg/kg per os) were used in all cases, including pulse therapy in two patients. Non-biologic DMARDs were prescribed in all cases, IVIG in 1/4 (25%), and 3/4 (75%) were treated using biologics (one—etanercept, one—tocilizumab, and one—adalimumab). All patients are still alive, and all had convincing positive dynamics, but still use medication. Patients 1 and 4 underwent surgery. Patient 4 had central nervous system (CNS) damage and sensorineural hearing loss. The clinical cases of COVID-19-associated vasculitis are described below.

Outcomes: Three patients now are in complete remission (1, 2, and 4), and patient 3, but still receives corticosteroids. Patient 1 is corticosteroid free and receives only adalimumab and warfarin, patient 2 receives etanercept monotherapy, and patient 4 is now free of any medications. A brief description of all above-mentioned cases is shown in Table 1.

## 4. Discussion

Our study describes the polymorphism of vasculitis associated with COVID-19. We described patients who developed vasculitis with and without previously diagnosed COVID-19, and with and without antibodies to SARS-CoV-2, which made diagnosis difficult. Tissue real-time PCR helped us investigate the possible SARS-CoV-2 etiology of vasculitis. The primary question regarding the role of SARS-CoV-2 in the pathogenesis of COVID-associated vasculitis remains unclear: is it a trigger of the known rheumatic disease or it is direct virus endothelial damage with attempts to remove the viruses from infected endothelial cells with subsequent immune dysregulation and vascular inflammation? In our case series, biopsies were performed during the disease course a maximum of 214 days after COVID-19 and 122 days after the initial presentations; therefore, our cases confirmed the long persistence of the viruses in the tissue (all patients had negative swab PCRs at the moment of biopsy). Theoretically, such long virus persistence may indicate the immune system’s failure to clear the infected cells of the virus. A similar inability to eliminate viruses was observed in many cases of virus-associated hemophagocytic lymphohistiocytosisbefore the COVID-19 pandemic as well as in multisystem inflammatory syndrome, where the persistence of the virus in the intestine set up hyperproduction of cytokines. Several studies confirmed prolonged virus isolation in stool despite the absence of the virus in the upper airways [10,11,12,13]. One suggested mechanism of vasculitis development is virus fixation through ACE2 receptors, a large number of which are located in the skin’s vascular endothelium. This activates the complement pathway, and the coagulation cascade leads to endothelial damage and expression of cytokines by the endothelium, which causes a cytokine storm [14].

Skin manifestations associated with COVID-19 are no longer new. Seven skin biopsies from pediatric patients (four male, three female; 11–17 years old) with skin vasculopathy were evaluated. Patients had skin lesions of the lower extremities similar to chilblain. Three patients had itching, and one had a slight pain syndrome. Some authors named these features chilblain or “COVID toes”. Similar to our study, all patients had negative nasopharyngeal and oropharyngeal PCR for SARS-CoV-2. The reliable contacts or signs of respiratory or dyspeptic syndrome were noted in less than 50% of cases. Patients had no underlying immune-mediated diseases, but two of them were treated with methylphenidate hydrochloride for attention deficit hyperactivity disorder. Histopathological examinations described lymphocytic vascular infiltration with signs of vasculitis and disruption ranging from endothelial swelling and endotheliitis to fibrinoid necrosis and thrombosis. The SARS-CoV-2 spike protein was most often found in the derma’s endothelial cells and the eccrine glands’ epithelium by immunohistochemical assays. The structures, morphologically similar to SARS-CoV-2, were found in the cytoplasm of endothelial cells via electron microscopy [15]. Negative results of standard COVID-19 diagnostics using nasopharyngeal PCR swabs may be the reason for the underdiagnosis of COVID-19 as a trigger or etiological factor of some vasculopathy. Identification of the SARS-CoV-2 virus in the tissues might be broadly recommended for patients with vasculitis, especially those patients with an unclassical course [15].

German researchers described the clinical case of an 81-year-old female patient with signs of COVID-19 infection and a macular rash with partial vasculitis-like patterns and predominant localization on the palms and soles. Again, standard PCR tests using upper respiratory swabs were negative. Skin PCR testing showed a low copy number of SARS-CoV-2 (37 per 1 × 10^6^ human RPPH1 copies) [16].

The overview of studies on skin lesions in children with COVID-19 included data from 119 studies, which showed that some skin manifestations were observed in only 6/2445 (0.25%) pediatric patients [6]. Disorders such as chilblain, maculopapular rash, erythema multiforme, and livened patterns were described. Skin biopsies for children with confirmed or suspected COVID-19 were rarely performed [6]. The authors noted that epidemiological history was important for the association between skin lesions and COVID-19, due to the low specificity of serological and nasopharyngeal swab PCR tests in children [6].

Thrombotic events remain an important effect of COVID-19 in adult and pediatric patients. Macro- and microvascular thrombosis of arteries, veins, arterioles, capillaries, and venules of internal organs were found in patients who died of COVID-19 [7]. It is known that normal vascular endothelium regulates the coagulation process and might be the source of proinflammatory cytokines [2,17]. The following pathogenetic mechanism of inflammation in the endothelium during COVID-19 has been assumed: endothelial cells produce various bioactive substances in response to damage, in particular cytokines, thrombin, and complement 5a. The NLRP3 inflammasome and the complement are stimulated, which leads to inflammation in the damaged blood vessels [2,18,19]. Based on this knowledge of the pathogenesis, new targets of treatment have been proposed, in particular, immune-mediated therapy with NLRP3-inflammasome antagonists, such as anti-IL-1, anti-IL-6, anti-GM-CSF, and colchicine [3,20,21].

Skin involvement might be associated with internal organ vasculitis. Multiple types of exanthema are described in the literature, including urticarial, petechial, and varicella-like rash with central nervous system vasculopathy with anti-myelin oligodendrocyte glycoprotein antibodies in patients with COVID-19 [2]. Clinically, vasculitis is more probable in children and adolescents, with skin lesions on the toes, feet, and hands, and with histological signs of dermatitis and vascular degeneration of the basal epidermal layer, endotheliitis, and lymphocytic infiltration [6]. Inflammatory infiltrates were detected via electron microscopy. They predominantly consisted of mature T cells and had positive staining on the SARS-CoV-2 spike protein in capillary endothelial cells with an immunohistochemical assay, as well as coronavirus-like particles [2]. Biopsies showed the presence of T cells with a CD4:CD8 ratio of approximately 3:2, a small number of B-lymphocytes and plasma cells, scattered CD30+-activated T cells, and plasmacytoid dendritic cells [22]. Vascular deposits containing IgM, IgA, and C3 with IgA antibodies against the SARS-CoV-2 spike protein S1 domain were observed in children with chilblain, suggesting that the immune inflammation was part of the pathogenesis [2,6,22]. In patients with severe COVID-19, an impaired interferon (IFN) type I response was observed [4]. Increased production of IFN-1 in patients with chilblain is aimed at attenuating viral replication. However, this strong and early response can induce microangiopathology changes similar to those in systemic lupus erythematosus (SLE) [4,23]. Video capillaroscopy showed pericapillary edema, dilated and abnormal-looking capillaries, and microhemorrhages in patients with COVID-19-associated vasculopathy, which were similar to those in SLE patients [2]. An Austrian cross-sectional study showed that higher values of dimethylarginine, von Willebrand factor, homocysteine, and β-2-glycoprotein antibodies, and lower levels of homoarginine and tryptophan in post-COVID-19 patients were associated with microangiopathy development (*p* < 0.05). These parameters of endothelial dysfunction were similar to patients with atherosclerosis [5]. A detailed pathogenesis of COVID-19-associated vasculopathy and one of the biggest case series were recently provided [24,25]. A systematic review of COVID-19–associated immunoglobulin A vasculitis containing information about 13 published cases with strictly male predominance (n = 12) was recently published [26].

The study limitations are related to the retrospective type of the series, the small number of observations, age and gender discrepancies, the possibility of incidental occurrence, and the absence of a control group.

## 5. Conclusions

COVID-19 is not just a new respiratory infection that causes severe lung damage. Detailed studies of the pathogenesis require a better understanding of COVID-19 mechanisms of damage to vital organs, such as blood vessels, the heart, lungs, and brain. Endothelial dysfunction, which has already been found in young patients, is perhaps a new and still underestimated challenge. High-quality and detailed diagnostics are important, especially biopsies of altered areas and searches for SARS-CoV-2 RNA, which shed some light on the pathogenesis of different types of vasculitides particularly associated with COVID-19. Information regarding clinical phenotypes, treatment efficacy, and outcomes is required to improve the management of such patients in the future and prevent fatal or serious disease consequences.

## Figures and Tables

**Figure 1 biomedicines-11-00870-f001:**
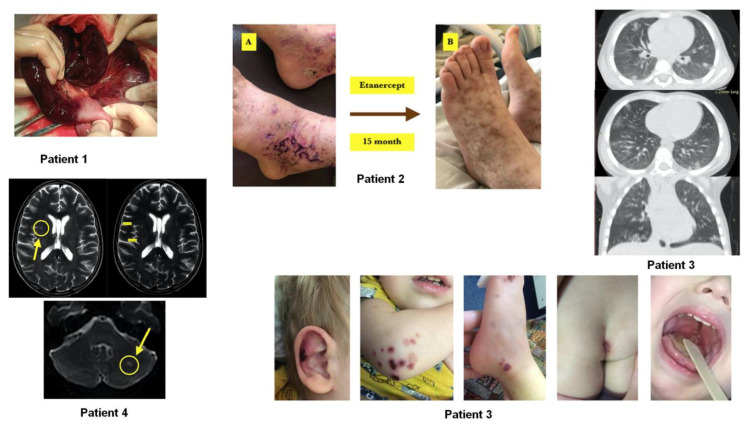
(1) Intraoperative pictures of patient 1: mesenteric vein thrombosis, intestine necrosis, and hemorrhagic peritonitis. (2) Skin manifestations in patient 2 before treatment (**A**) and after treatment (**B**). (3) Ischemic changes in patient 3 and scan of patient 3’s chest organs had shown areas of reduced pneumatization and enhanced density with ground-glass opacities. (4) Brain MRI of patient 4. Arrows show hyperintense foci in the cerebellum with cytotoxic edema (in the bottom) and areas of ischemic infarction in the white matter (upper part).

**Table 1 biomedicines-11-00870-t001:** Clinical and laboratory characteristics of four patients with vasculitis associated with SARS-CoV-2, types of therapy, and outcomes.

ID	Sex (m/f)/ Age (y)	Previous COVID-19 (Y/N)	SARS-CoV-2 Identification (sPCR, IgM, IgG)	Type of Vasculitis	Involved Tissues	Biopsy	Systemic Inflammation/ Lab Results				Treatment	Outcomes	Ct Values for SARSCoV-2 Genes in Affected Tissues			
							ESR (mm/h)	CRP (mg/l)	Ferritin	D-dimer			ORF1	ORF8	N	Result
1	M/16	Y	IgG	UV	Lungs, heart, GI, TE	jejunum	25	24	158	37,018	GCS, LWH, ADA	CR	31.6	30.5	32.0	POS
2	F/15	Y	sPCR, IgG	PA	Skin necrosis	skin	15	4	17	237	GCS, MTX, ETN	CR	31.1	31.2	31.6	POS
3	M/2	N	IgM, IgG	IgAV/HSP	Skin necrosis, lungs, heart, GI	stomach	16	23	27	2808	GCS, CTX→MMF	PR	32.2	30.5	32.4	POS
4	M/15	N	N	PA	Skin, retina, lungs, kidney, inner ear, brain, heart, GI	stomach	23	153	432	1288	GS, LWH, HCQ, TCZ, IVIG	CR	28.6	27.5	30.1	POS

Abbreviations: ADA—adalimumab, CR—complete response, CTX—cyclophosphamide, ETN—etanercept, GCS—glucocorticosteroid therapy, GI—gastrointestinal system, HCQ—hydroxychloroquine, IgAV/HSP—IgA vasculitis/Henoch–Schonlein purpura, LWH—low-weighted heparin, MMF—mofetil mycophenolate, MTX—methotrexate, N—no, NEG—negative, PA—polyarteritis nodosa, POS—positive, PR—partial response, sPCR—RT-PCR from swabs, TE—thrombotic events, TCZ—tocilizumab, UV—unclassified vasculitis, Y—yes.

## Data Availability

Data are unavailable due to privacy or ethical restrictions.

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
