# Peer review of "COVID-19 Associated Vasculitis Confirmed by the Tissues RT-PCR: A Case Series Report"

_biomedicines, 2023, doi:10.3390/biomedicines11030870_

Round 1

Reviewer 1 Report

The paper is very interesting. It is well-organized, detailed, and easy to understand. However, as the limitations the authors point out, the small sample size and without a control group make the results not very reliable. Also, the authors do not mention whether these four patients have any other diseases before they got COVID-19 that can cause vasculitis. Moreover, there is only one female. Is that reasonable to be comparable to the other three males, that is, gender disparity? Also, there is one patient aged two years old, much younger than the other three patients. Is there any possible age disparity or is it because of the potential diseases under his younger age rather than because of COVID-19?

Author Response

Dear Reviewer!

Thank you so much for you kind evaluation. Inded, the small sample size is the main problem of our manuscript and the gender and age discrepancy confirmed the bias. Comparing to predisposition of elder age in MIS-C children supposed to elder age of COVID-19 associated vasculopathies too. I think younger age might be related with specofoc type of vasculitis- Ig A vasculopathy. Our patients did not have any previous health problems, which may cause vasculitis. In all patient the whole exome patients did not reveal any relevant diseases, e.g. primarily immunodeficiency or monogenic vasculitis, e.g. DADA2. The information about these point were added in the manuscript and highlighted with green colours.

Sincerely yours

Mikhail Kostik, MD, PhD, Professor

Reviewer 2 Report

Authors have done wonderful study. However, following things should addressed to improve the quality of the work:

-At least two more samples should be added to make the study statistically significant. What about control samples?

-What could be effect of COVID-19 vaccination in these patients?

-Rectify the typographical mistakes from the entire manuscript. Some examples here:

In materials and methods section, authors have mentioned “Materials and Methods In the retrospective-prospective case, series study” while it should be “…. Retrospective-prospective case series study,…” “comma” misplaced…

Similarly in the same section: point no 2, authors mentioned that “positive SARS-CoV –“ it shoudl be ”positive SARS-CoV -2”

Author Response

Dear Reviewer!

Thank you so much for you very positive evaluation of our manuscript.

Our answers (A) to your query (Q) are below. All changes in the manuscript are highlighted with green colour.

Authors have done wonderful study. However, following things should addressed to improve the quality of the work:

Q1)-At least two more samples should be added to make the study statistically significant. What about control samples?

A1) Dear Reviewer, our database includes more than 1000 children with COVID-19 and couples of dozen vasculitis. I know about similar cases in different cities (one patient with mesenterial thrombosis, one young child with IgA vasculits with severe lung involvement and one with arterial thrombosis). Theoretically we can ask colleagues to give us new cases, but it required more time for assessment and include co-authors. I think this question required to be asked from editorial office. So I am waiting your suggestion. The tissue PCR SARS-CoV-2 confirmation had the last patient, but we did not include him due to different phenotype and like thrombophilic disease. Other known cases have clinical and serological SARS-CoV-2 confirmation but has not had tissue PCR. I think in future we can try to make tissue PCR in other vasculitis cases despite the information if they were SARS-CoV-2 positive or now.

The PCR assessment was performed with positive and negative controls during the PCR tests by our geneticists.

Q2)-What could be effect of COVID-19 vaccination in these patients?

A2) This is a very challenging question. In one hand additional dose of

antigen theoretically may again activate the immune system, but I think during the treatment with biologics and anticoagulants it may be safe. From other hand I think the vaccination may decrease the incidence of such cases, so further trials are required.

Q3)-Rectify the typographical mistakes from the entire manuscript. Some examples here:

In materials and methods section, authors have mentioned “Materials and Methods In the retrospective-prospective case, series study” while it should be “…. Retrospective-prospective case series study,…” “comma” misplaced…

Similarly in the same section: point no 2, authors mentioned that “positive SARS-CoV –“ it shoudl be ”positive SARS-CoV -2”

A3) Dear Reviewer!

Thank you so much again. The manuscript was re-evaluating and typos were improved.

Thank you so much!

Mikhail M. Kostik, MD, PhD, Professor

Reviewer 3 Report

In this clinical case report four pediatric cases of COVID-19-associated vasculitis are presented, collected in a retrospective-prospective manner.

A recent systematic review on this subject (doi:10.1016/j.jen.2022.05.002) could be added.

It is indeed a rare but life-threatening complication especially for children. This report is important for its description of treatments and outcomes.

The table describes well the clinical characteristics of the patients (it erroneously mentions five instead of four cases). The figure shows the thrombotic lesions assessed operatively, macroscopically and through MRI. The quality is good.

Author Response

Dear Reviewer!

Thank you so much for you very positive evaluation of our manuscript.

Our answers (A) to your query (Q) are below. All changes in the manuscript are highlighted with green colour.

In this clinical case report four pediatric cases of COVID-19-associated vasculitis are presented, collected in a retrospective-prospective manner.

Q1) A recent systematic review on this subject (doi:10.1016/j.jen.2022.05.002) could be added.

A1) Dear Reviewer! The recommended manuscript cited in the text and reference list.

Q2) It is indeed a rare but life-threatening complication especially for children. This report is important for its description of treatments and outcomes. The table describes well the clinical characteristics of the patients (it erroneously mentions five instead of four cases). The figure shows the thrombotic lesions assessed operatively, macroscopically and through MRI. The quality is good.

A2) Dear Reviewer! The error was changed. Thank you so much for your positive mark of our work.

Thank you so much!

Mikhail M. Kostik, MD, PhD, Professor